# Stimulus dependencies—rather than next-word prediction—can explain pre-onset brain encoding in naturalistic listening designs

Inés Schönmann[1]*, Jakub Szewczyk[1,2], Floris P de Lange[1], Micha Heilbron[1,3]

[1]Donders Institute for Brain Cognition and Behaviour, Nijmegen, Netherlands; [2]Institute of Psychology, Jagiellonian University, Kraków, Poland; [3]Amsterdam Brain and Cognition, University of Amsterdam, Amsterdam, Netherlands

## eLife Assessment

This **fundamental** study investigates whether neural prediction of words can be measured through pre-activation of neural network word representations in the brain; **convincing** evidence is provided that neural network representations of neighboring words are correlated in natural language. This study urges future studies to carefully differentiate between neural activity that predicts the upcoming word and neural activity that encodes the current words, which contain information that can be used to predict the upcoming word. The study is of potential interest to researchers investigating language encoding in the brain or in large language models.

*For correspondence:
ines.schoenmann@gmail.com

**Abstract** The human brain is thought to constantly predict future words during language processing. Recently, a new approach emerged that aims to capture neural prediction directly by using vector representations of words (embeddings) to predict brain activity prior to word onset. Two findings have been proposed as hallmarks of neural next-word prediction: (i) significant encoding prior to word onset and (ii) its modulation by word predictability. However, natural language is rife with temporal correlations, where upcoming words share statistical information with preceding ones. This raises a critical question: Do these hallmarks emerge from the brain actively predicting future content, or might they be equally well explained by the regression model exploiting these inherent stimulus dependencies? To distinguish between these alternatives, we applied the same encoding analysis to passive control systems, i.e., representational systems that encode the stimulus but cannot predict upcoming words. We show that both hallmarks emerge in two such control systems, namely in word embeddings themselves and in speech acoustics. We further show that proposed methods to correct for these dependencies are insufficient, as the effects persist even after such corrections. Together, these results suggest that pre-onset prediction of brain activity might reflect dependencies in natural language rather than predictive computations. This questions the extent to which this new encoding-based method can be used to study prediction in the brain.

## Introduction

In the past years, the field of natural language processing (NLP) has made great advances in developing computational systems that can generate, classify, and interpret language. Much of this progress has been driven by large language models (LLMs): neural networks trained in a self-supervised

manner to predict the next word or token (*Minaee et al., 2024*). Surprisingly, this simple training objective is sufficient for models to learn about language more broadly, making models develop a human-like knowledge of syntax (*Manning et al., 2020*; *Linzen and Baroni, 2021*) and enabling them to solve almost any NLP task (*Minaee et al., 2024*; *Manning, 2022*). Furthermore, 'embeddings,', also constitute the highest performing encoding models for predicting brain responses to linguistic stimuli (*Caucheteux and King, 2022*; *Jain and Huth, 2018*; *Schrimpf et al., 2021*)—indicating that LLMs' internal representations might capture some aspects of human language representations (*Tuckute et al., 2024*).

Two lines of research suggest that predicting the upcoming linguistic input also plays an important role in *human* language comprehension. The first line of research focuses on neural and behavioural responses occurring *after* the onset of the stimulus in question. The reasoning here is that if the brain is engaged in predicting upcoming linguistic input, brain responses, and reading times should vary as a function of linguistic predictability (*Smith and Levy, 2013*; *Frank et al., 2015*; *Willems et al., 2016*; *Shain et al., 2024*; *Heilbron et al., 2023*). Many studies following this line of reasoning have demonstrated that both neural responses and reading times are sensitive to even subtle fluctuations in predictability at several levels of linguistic analysis, e.g., phonemes, words, or semantics (*Boston et al., 2011*; *Brodbeck et al., 2022*; *Heilbron et al., 2022*; *Smith and Levy, 2013*; *Szewczyk and Federmeier, 2022*). This approach can be seen as an extension of a long-standing tradition of research into linguistic expectations which relied on carefully constructed sentences that violate linguistic expectations and use cloze probabilities to quantify unexpectedness (*Kutas and Hillyard, 1980a*; *Kutas and Hillyard, 1980b*, *Kutas and Hillyard, 1984*).

Recently, a new, alternative approach emerged that aims to probe linguistic predictions directly by capturing their neural signature *prior* to the onset of a word (*Wang et al., 2018*; *Goldstein et al., 2022b*). Predicting a word is thought to involve pre-activating the representation of that word. Hence, finding a trace of a representation of a word in the neural signal *prior* to its onset is interpreted as direct evidence for a word's pre-activation, and therefore, next-word prediction. Capturing the neural signature of the prediction itself is appealing, as it has the potential to circumvent interpretational challenges of more indirect, post-stimulus predictability effects which have been suggested to reflect related but distinct downstream processes—such as semantic integration difficulty or 'postdiction'—rather than prediction per se (*Pickering and Gambi, 2018*; *Huettig, 2015*; *Huettig and Mani, 2016*). Indeed, one of the most widely used post-stimulus measures of prediction, surprisal, was originally proposed as a measure of syntactic integration difficulty and not as a measure of prediction (*Hale, 2001*). Perhaps the most influential demonstration of predictive pre-activation during language comprehension was presented by *Goldstein et al., 2022b*. Using encoding models on electrocorticographic (ECoG) recordings of participants listening to naturalistic speech, they reported two findings which we will refer to as *hallmarks of prediction*: (i) brain responses could be predicted significantly better than chance as early as 2 s prior to word onset, and (ii) this pre-onset encoding was modulated by word predictability, with highly predictable words showing stronger pre-onset encoding. This modulation by predictability is in line with the idea that their representation was pre-activated more strongly. More recently, a similar pattern of results was found in non-invasive MEG recordings (*Azizpour et al., 2024*).

However, interpreting pre-onset encoding as evidence for prediction is not as straightforward as it may seem. This is because language is rife with temporal dependencies which can potentially be learned by a regression model: Neighbouring words often share semantic content—as in 'pine tree' or 'green leaves'—or morphosyntactic features, as in 'he goes' where both words carry syntactic markers denoting third person singular. Other dependencies are not structural but incidental, such as those caused by words that happen to co-occur together often, such as 'Sherlock Holmes.' Irrespective of the exact nature of these dependencies, they allow to predict (at least in part) earlier words from subsequent ones. Therefore, they might also allow to predict (at least in principle) brain responses to earlier words using word representations of subsequent words. A priori, then, these inherent stimulus dependencies might already explain why representations of future words can be used to model prior brain responses, without having to assume any predictive pre-activation by the brain. This creates a fundamental ambiguity: Is pre-onset encoding a reflection of the brain generating predictions about upcoming words, or does it merely show that the regression model can successfully exploit temporal dependencies present in the stimulus material?

One way to distinguish between these alternatives is to use what we call a *passive control system*: a representational system in which the stimulus—and thus its dependencies—are encoded but that, by definition, cannot generate predictions about upcoming words. Speech acoustics provide such an example, as the auditory stimulus is encoded in the speech acoustics, yet they cannot actively 'predict' upcoming words. If, when applying the same analysis to such a control system, the hallmarks of prediction still emerge, this would demonstrate that they must arise from stimulus dependencies alone, without requiring any underlying predictive process.

Here, we directly address this issue. First, we replicate the results reported by *Goldstein et al., 2022b* and *Azizpour et al., 2024* across two magnetoencephalography (MEG) datasets, demonstrating that both hallmarks robustly generalise to MEG. We then evaluate two passive control systems—word vectors and speech acoustics—neither of which actively predicts upcoming words. We show that both purported hallmarks emerge in these control systems, despite the absence of any predictive process. Furthermore, we demonstrate that methods proposed to correct for stimulus dependencies, such as removing reoccurring bigrams or residualising neighbouring word information from embeddings, prove insufficient: The proposed hallmarks persist in the acoustic control system even after such corrections. We conclude that both proposed hallmarks can be fully explained by the correlational structure inherent in naturalistic language, without assuming predictive pre-activation in the neural data. This poses a challenge to the use of encoding models for probing linguistic predictions and questions to what extent such analyses can demonstrate that brains, like LLMs, perform next-word prediction.

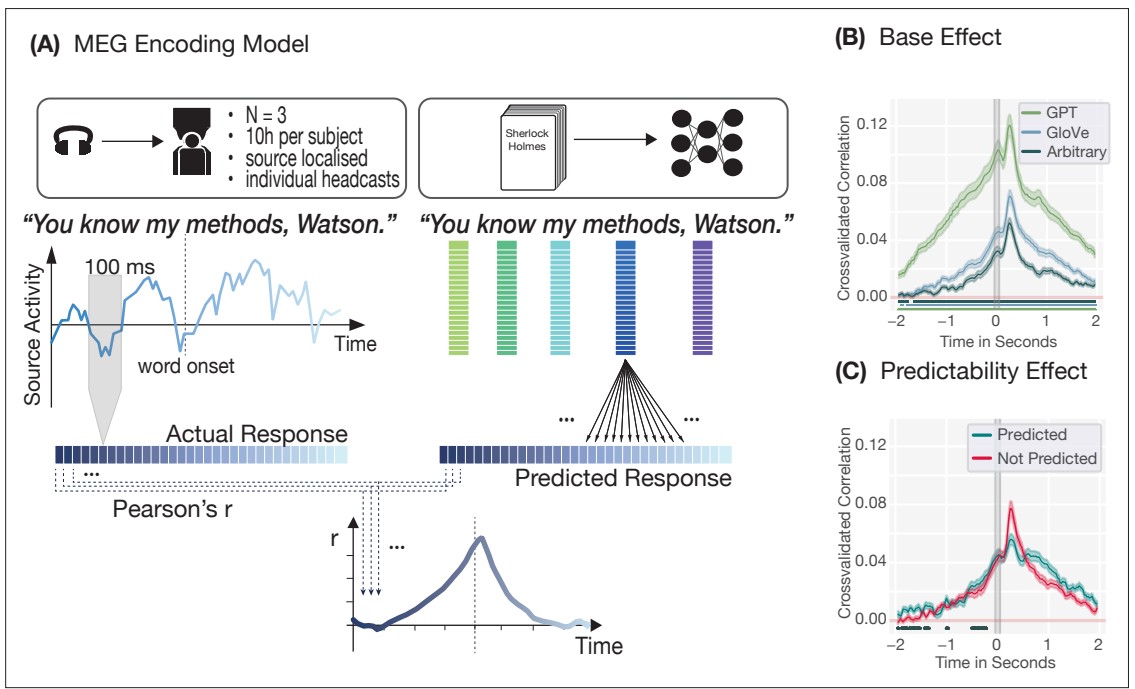

**Figure 1.** MEG encoding model and results for encoding neural data. (**A**) Magnetoencephalography (MEG) encoding model. MEG data was epoched to word onset and averaged over a sliding window of 100 ms, moving with a step size of 25 ms. The model representation (GPT-2, GloVe, or arbitrary) of the word at $t = 0$ was then used to predict the neural response for each channel and time point in a separate cross-validated Ridge regression. The actual and predicted responses were then correlated time point by time point, resulting in a time-resolved encoding plot. (**B**) Positive pre-onset encoding (subject 1) for GPT-2 (green), GloVe (blue), and arbitrary (grey) embeddings shows that it is possible to find ostensible neural signatures of pre-activation in MEG data. Lines show clusters of time points that are significantly different from zero ($p<0.05$ under the permutation distribution). (**C**) Encoding using GloVe embeddings demonstrates a slight advantage of the predictability of a word (top-one prediction by GPT-2-XL) for pre-onset encoding. The line indicates clusters of time points prior to word onset during which predictable words are significantly better encoded ($p<0.05$ under the permutation distribution).

The online version of this article includes the following figure supplement(s) for figure 1:

**Figure supplement 1.** Hallmarks of prediction in the remaining two subjects of the few-subject dataset and the multi-subject dataset.

# Results

## Hallmarks of prediction replicate in MEG data

Drawing on two different, publicly available MEG datasets in which participants listened to narratives, we analysed the data following the approach put forth by *Goldstein et al., 2022b* and *Azizpour et al., 2024*. Hence, MEG data were epoched with respect to the onset of each word between $-2s$ and $+2s$, and brain activity was averaged over a sliding window of 100 ms. Subsequently, the word representation (word embedding) corresponding to the word to which each epoch was time-locked was used to predict brain activity within that epoch, i.e., within the time window of $-2s$ to $+2s$ (see *Figure 1a*). Correlating the predicted with the actual brain response results in a time-resolved prediction accuracy curve which allows us to test the two hallmarks of prediction proposed by *Goldstein et al., 2022b*, namely (i) whether brain activity prior to the onset of a word can be predicted from that word's embedding and (ii) whether pre-onset prediction accuracy is higher for predictable words than for unpredictable ones. These two aspects are proposed as evidence of prediction, since pre-onset encoding is thought to capture the pre-activation of a word's neural representation.

Using representations from three different models (GPT-2, GloVe, and arbitrary 300-dimensional word embeddings) to encode brain activity, we found that it is possible to replicate both hallmarks of predictions described by *Goldstein et al., 2022b*. We found both (i) positive encoding prior to word onset for all three models (see *Figure 1B*) and (ii) a slight encoding advantage prior to word onset for highly predictable (i.e. GPT-2's top-one prediction) as opposed to less predictable words (see *Figure 1C*). For subject 1, this advantage started as early as 575 ms prior to word onset, and predictable words led to a continuously higher encoding performance (M=0.004, SD=0.001, p<0.031), corresponding to an average improvement in encoding performance within this pre-onset time window of 17% with respect to unpredictable words (with improvements of M=.006 (SD=0.002, p<0.035) corresponding to 25% starting 525 ms prior to onset, and M=0.011 (SD=0.006, p<0.033) corresponding to 46% during the time window of 1400-300 ms prior to word onset for subjects 2 and 3, respectively, see *Figure 1—figure supplement 1A and B*). In accordance with previous findings, using GPT-2's contextualised word representations allowed for earlier and better brain encoding than using non-contextualised GloVe embeddings (*Goldstein et al., 2022b*; *Schrimpf et al., 2020*). In turn, these non-contextualised word embeddings predicted brain response better than arbitrary, 300-dimensional vectors. However, arbitrary embeddings still performed remarkably well, given that they contain no structured information besides the identity of a word.

These results replicated for all three participants (see *Figure 1—figure supplement 1A and B*) in the few-subject dataset, indicating the robustness of the results given sufficient amounts of data (see *Figure 1—figure supplement 1C* for results from a more conventional multi-subject dataset, and see discussion section for possible explanations for this discrepancy). This demonstrates that pre-onset encoding is a tremendously robust phenomenon, replicating across different word embeddings (GPT-2, GloVe, and even arbitrary embeddings), datasets (single-subject as well as multi-subject) and even different forms of MEG spaces (source as well as sensor data). This suggests that pre-onset encoding accuracy is driven neither by the specific neural data nor by the specific word representations used in the encoding model, further raising the question to which extent the stimulus itself might be driving the effect.

## Both hallmarks emerge in passive control systems

Having established that both purported hallmarks generalise to MEG data, we next ask to what extent they can unequivocally be interpreted as reflecting neural pre-activation. For this question, we turn to the control systems: In order for these hallmarks to serve as evidence for signatures of neural predictive processes, they would have to be unique to brain responses. If, however, they can arise from stimulus dependencies alone, they should also appear when applying the same analysis to control systems, i.e., systems in which the stimulus is encoded but which cannot generate predictions.

The first control system we considered consisted of the word embeddings themselves, namely vector representations of the current word in which linguistic information is encoded but which do not perform any predictive computation. We applied the identical encoding analysis used for modelling the MEG data, but replaced the neural signal with word embeddings at each time point (*Figure 2A*). The embedding at $t = 0$ was used to predict embeddings at earlier time points. The degree to which an encoding model can predict earlier embeddings from later ones—which we call

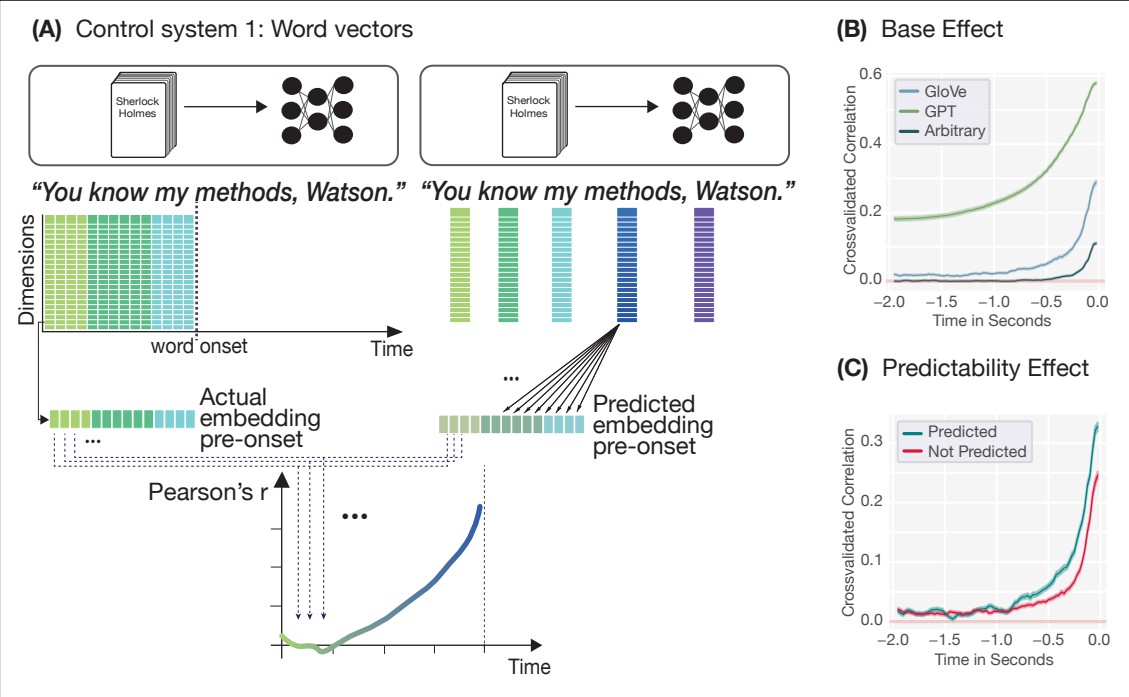

**Figure 2.** Control system 1: encoding model and results. (**A**) Control system 1: word embeddings. For the first control system, we performed the same analysis as in *Figure 1* but replaced the neural data at each time point with a vector representation (embedding) of the word presented at that time point. The word vector at $t = 0$ was then used to predict the previous word vector for each dimension and time point in a separate cross-validated Ridge regression. The actual and predicted values were then correlated time point by time point, resulting in a time-resolved self-predictability plot. (**B**) Pre-onset encoding (*self-predictability*) for GPT-2 (green), GloVe (blue) and arbitrary (grey) embeddings. Shaded areas show 95 % confidence intervals computed over model dimensions. (**C**) Modulation of pre-onset encoding of static (GloVe) word embeddings by contextual predictability: Prior word vectors are better predicted by successive word vectors if the subsequent word is highly predictable in context (i.e. top-one prediction by GPT-2). Shaded areas show 95 % confidence intervals computed over model dimensions.

The online version of this article includes the following figure supplement(s) for figure 2:

**Figure supplement 1.** Self-predictability in the multi-subject dataset and in *Goldstein et al., 2022b* data.

*self-predictability*—reflects temporal dependencies in the word vector space itself. We tested three types of embeddings: GPT-2 representations, static GloVe vectors, and arbitrary vectors containing no linguistic structure beyond word identity. Note that GPT-2 embeddings are not actually a passive control since these embeddings are the internal representations of a model that is actively predicting the next word. The critical test in this analysis is whether the hallmarks emerge for static and arbitrary embeddings. Nevertheless, we include GPT-2 embeddings for completeness and for comparability to the neural results. We observe that our first control system exhibits both hallmarks of prediction (*Figure 2B–C*): Pre-onset encoding was significantly above chance for all three vector spaces, and this effect was modulated by word predictability (for a replication of these results for the multi-subject dataset and the stimuli used by *Goldstein et al., 2022b* see *Figure 2—figure supplement 1*). In other words, preceding word vectors were predicted better when the subsequent word was highly predictable in context—mirroring the pattern observed in neural data, but emerging from stimulus structure alone.

Importantly, however, our self-predictability analysis did not require a mapping between different representational spaces, as is the case when encoding neural data, where word embeddings are used to predict brain responses. Hence, we next tested whether the same hallmarks emerge when using word embeddings to predict another meaningful, passive control system, namely the speech acoustics. Like in the case of word embeddings, the stimulus is encoded in the acoustics, yet the acoustics themselves do not perform any active prediction of upcoming words. Furthermore, since participants listened to the narratives, a representation of the speech acoustics must be present in the neural representation. Consequently, the speech acoustics constitute a meaningful, passive control system when testing for the influence of stimulus dependencies on encoding results.

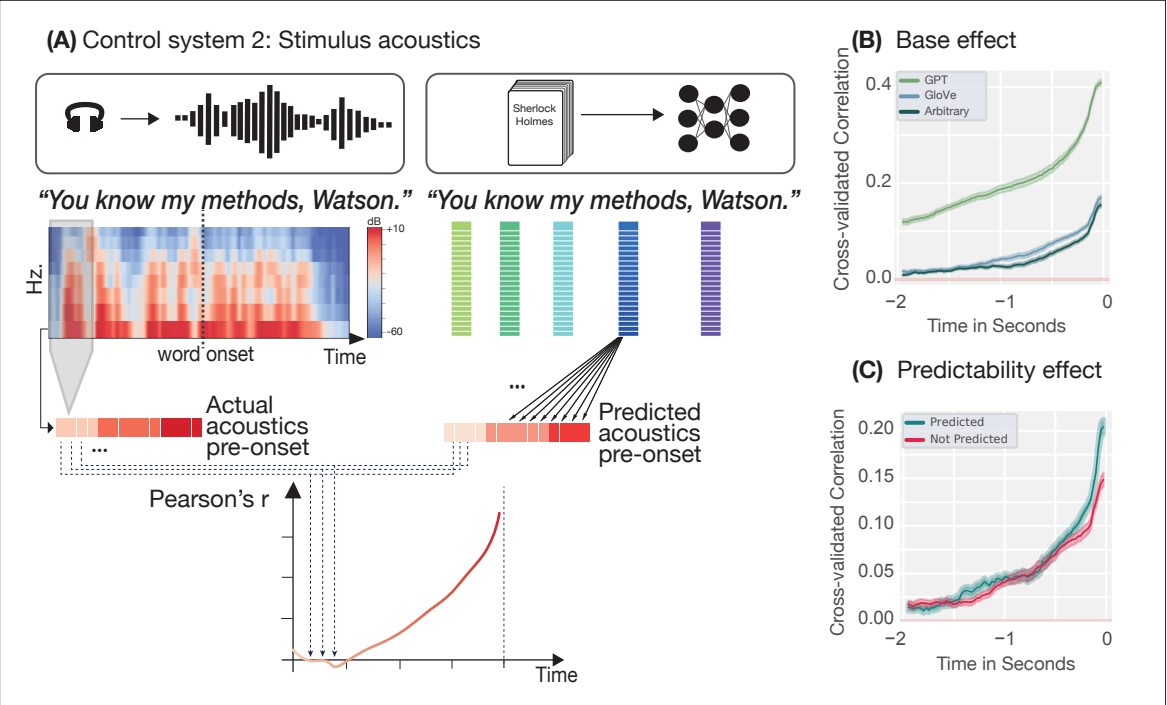

**Figure 3.** Control system 2: encoding model and results. (**A**) For the second control system, we again perform the same analysis as in *Figure 1* but replaced the neural data with the stimulus acoustics at that time point. The word embedding of the word at $t = 0$ was then used to predict the prior acoustics. The actual and predicted values were then correlated time point by time point, resulting in a time-resolved correlation plot. (**B**) Pre-onset encoding of speech acoustics based on GPT-2 (green), GloVe (blue), and arbitrary (grey), embeddings. Shaded areas show 95 % confidence intervals computed over the nine dimensions (8 mels + envelope). (**C**) Modulation of pre-onset encoding of speech acoustics based on GloVe embeddings by word predictability (top-one prediction by GPT-2-XL). Shaded areas show 95 % confidence intervals computed over the nine dimensions (8 mels + envelope).

The online version of this article includes the following figure supplement(s) for figure 3:

**Figure supplement 1.** Predicting acoustics prior to word onset from *original* embedding vectors for both datasets.

**Figure supplement 2.** Predicting acoustics from arbitrary embedding vectors for all three datasets after all reoccurring bigrams have been removed.

**Figure supplement 3.** Encoding data using GPT, GloVe, and arbitrary vectors after removing re-occurrences of bigrams, i.e., only retaining the first occurrence, in our few-subject dataset.

We extracted acoustic features (an 8-band Mel spectrogram and envelope) for each word and applied the same encoding analysis, using the GPT-2, GloVe, or arbitrary embedding at $t = 0$ to predict acoustic features at earlier time points (*Figure 3A*). Again, both hallmarks of prediction emerged in naturalistic speech acoustics (*Figure 3B–C*, and for a replication of these results for the data used by *Goldstein et al., 2022b* see *Figure 3—figure supplement 1B*). Acoustic features prior to word onset could be predicted from the embedding of the upcoming word in all three datasets (*Figure 3B*, *Figure 3—figure supplement 1*), and this effect was modulated by word predictability. In other words, pre-onset acoustics could be predicted better when the subsequent word was highly predictable. Note, however, that this modulation did not occur for the acoustic data of our multi-subject dataset, the speech of which was less naturalistic and for which we could compute only poorer acoustic word representations (see Methods/Discussion).

These findings demonstrate that the proposed hallmarks can emerge in control systems that, by definition, cannot predict upcoming words. Observing these hallmarks in neural data, therefore, does not, by itself, demonstrate that the brain is generating predictions: Hallmarks could instead equally reflect the regression model capitalising on stimulus dependencies which are passively encoded by the brain.

## Proposed controls do not effectively remove stimulus dependencies

The previous analyses show that both hallmarks of prediction emerge in passive control systems, indicating that stimulus dependencies alone—rather than next-word prediction—may explain effects found in the neural data. This quite naturally raises the questions of whether such stimulus dependencies can be controlled for. A first control already proposed by *Goldstein et al., 2022b* was to test whether pre-onset encoding persists even when removing all trials containing reoccurring bigrams for arbitrary embeddings—i.e., random vectors containing no linguistic structure that are assigned to each word. Here, the logic is that nothing but word identity is encoded in arbitrary vectors. Hence, since neighbouring words share no systematic information beyond their incidental co-occurrence in a specific text, removing repeated bigrams eliminates this association. The authors argued that if pre-onset encoding persisted under such conditions, results should be driven by neural pre-activation rather than stimulus dependencies. However, when applying this same constrained analysis, not only did neural encoding remain above chance (see *Figure 3—figure supplement 3A*), but crucially, removing reoccurring bigrams did not influence pre-onset encoding in either of our two control systems (see *Figure 3—figure supplement 3B-C*). Specifically, across all three datasets, we observe the same results that have hitherto been presented as evidence against encoding stimulus dependencies in our acoustic control system (*Figure 3—figure supplement 2*): Arbitrary embeddings could predict pre-onset acoustics significantly above chance, hence encoding the stimulus even after bigram removal. These results demonstrate that this proposed control is insufficient. Removing reoccurring bigrams does not prevent the regression model from encoding stimulus dependencies. Instead, our results indicate that regression models can and do exploit dependencies that go beyond mere re-occurrences of bigrams to predict pre-onset features.

A more rigorous approach to removing stimulus dependencies is residualisation: regressing out neighbouring words from each embedding before using it as a predictor in the encoding analysis. For

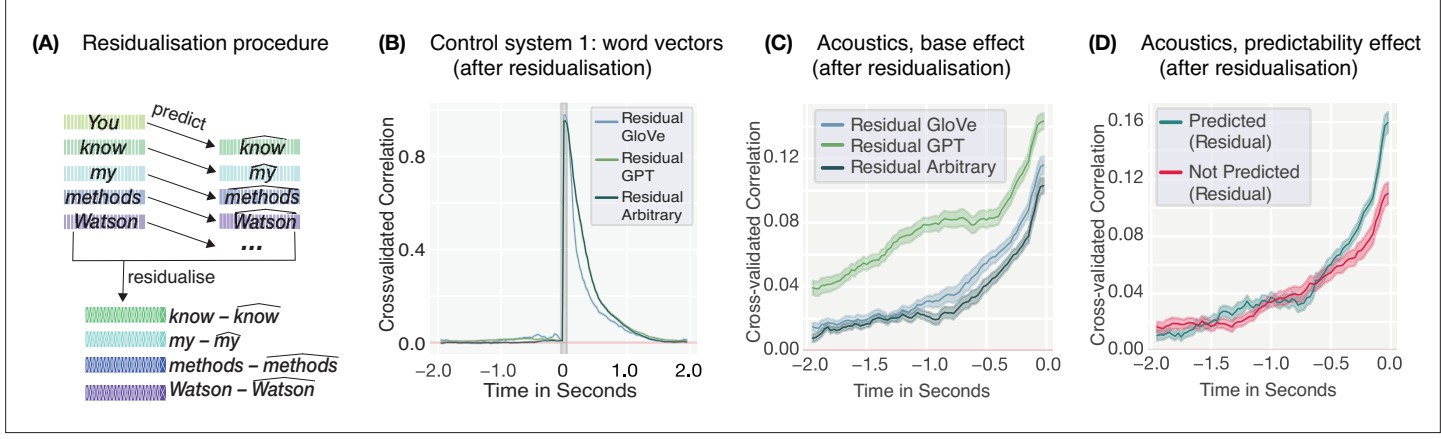

**Figure 4.** Controlling for self-predictability. (**A**) In order to remove shared information between a word and its predecessor in the text, we residualised word embeddings by first fitting an OLS regression to predict the next word based on the previous word's embedding, i.e., predicting 'know' based on 'You.' This resulted in a predicted embedding $\hat{x}$, e.g., '$\widehat{know}$,' which contained the shared information between the two words. Finally, the predicted embedding, e.g., '$\widehat{know}$,' was removed from the original embedding, e.g., 'know,' to generate word representations for which the dependency between neighbouring words was removed. (**B**) Self-predictability after regressing out the previous embedding from the embedding at $t = 0$ shows that it is possible to successfully remove the correlations between neighbouring model representations. For brain encoding results when using these residualised embeddings, see *Figure 4—figure supplement 1*. (**C**) Predictability of prior word acoustics when using residualised GPT-2 (green), GloVe (blue), and arbitrary (grey) embeddings prior to word onset. Patterns closely mirror those observed in each model's self-predictability or in residual brain encoding results. Shaded areas show 95 % confidence intervals computed over the nine dimensions (8 mels + envelope). (**D**) Predictability of prior word acoustics using residualised GloVe embeddings demonstrates a clear advantage of the predictability of a word (top-one prediction by GPT-2-XL) for predicting its prior acoustic representations, and therefore, the same qualitative difference as observed when encoding neural data. Shaded areas show 95 % confidence intervals computed over the nine dimensions (8 mels + envelope).

The online version of this article includes the following figure supplement(s) for figure 4:

**Figure supplement 1.** Predicting acoustics prior to word onset from *residual* word embeddings in the multi-subject dataset.

**Figure supplement 2.** Ostensible hallmarks of prediction after removing model self-predictability through residualising word embeddings in the few-subject dataset and the multi-subject dataset.

instance, in the sentence 'You know my methods,' 'You' can be regressed out of 'know,' 'know' out of 'my,' and 'my' out of 'methods' (see *Figure 4A* for a visualisation). We residualised each word embedding—thereby removing all information from that embedding that could be predicted linearly from its predecessor. Note that this is a generalised version of a control analysis performed by *Goldstein et al., 2022b* which removed neighbouring words through projection. When we recomputed self-predictability using these residualised embeddings, temporal dependencies were successfully eliminated: The encoding model could no longer predict earlier embeddings from later ones (*Figure 4B*). This confirms that residualisation effectively removes dependencies within the embedding space.

Strikingly, however, when we used these residualised embeddings to predict our acoustic control system, both hallmarks persisted, insofar as they were present in the original modelling effort (*Figure 4*, *Figure 4—figure supplement 2*). Pre-onset acoustics could still be predicted above chance, and this effect was still modulated by word predictability in both datasets that used natural speech, despite information about neighbouring embeddings being linearly removed. This reveals a fundamental limitation of the approach: Residualisation cannot account for stimulus dependencies when mapping *between* representational spaces. It only removes a very specific representation of neighbouring words from the embeddings, but it fails to remove the underlying dependencies that exist in language itself. However, these underlying dependencies are still present in the system that is modelled, i.e., in the acoustics or the neural data. As long as word embeddings still identify words, the regression model will be able to relearn these (cross-space) stimulus dependencies. Correcting the embeddings through residualisation is, therefore, insufficient: As long as the signal being predicted contains the temporal structure inherent in naturalistic language, the proposed hallmarks will emerge.

Finally, it is important to note that in order to ensure that these findings are not unique to one specific stimulus set, we applied the same control analyses to the stimulus material from the original study by *Goldstein et al., 2022b*, and consistently observed the same effects: Proposed hallmarks of prediction emerged in both control systems (*Figure 2—figure supplement 1B*, *Figure 3—figure supplement 1B*) and could not be corrected for by either bigram removal (*Figure 3—figure supplement 2C*) or residualisation (*Figure 4—figure supplement 1B*). Our results, therefore, demonstrate that the proposed hallmarks of prediction observed in neural encoding studies can be explained fully by stimulus dependencies, not only in the MEG datasets analysed here, but in the very dataset used to establish pre-onset encoding as evidence for neural prediction.

## Discussion

We evaluated an encoding modelling paradigm that uses vector representations of words to capture neural signatures of linguistic predictions during naturalistic listening. Specifically, we asked to what extent two proposed hallmarks of prediction—namely (i) positive encoding performance prior to word onset, and (ii) sensitivity to the predictability of a word—can be interpreted as reflecting neural pre-activation. Across two MEG datasets, we found that both hallmarks appear not only in neural data but also in two passive control systems: static word embeddings and the stimulus acoustics. Since these control systems encode the stimulus but cannot predict upcoming words, these results demonstrate that the proposed hallmarks can arise from stimulus dependencies alone, without assuming any predictive pre-activation in the brain. We further showed that methods proposed to correct for these dependencies—removing reoccurring bigrams or residualising neighbouring word information from embeddings—are insufficient, since pre-onset encoding persists in the acoustic control system even after such corrections. These results reveal a fundamental ambiguity in the approach. Both proposed hallmarks can be fully accounted for by stimulus dependencies alone, without assuming any prediction in the brain.

We observed that the first hallmark of prediction—pre-onset encoding—is tremendously robust, replicating not only in non-invasive MEG data (which has a lower signal-to-noise ratio compared to the ECoG data used by *Goldstein et al., 2022b*), but also across various types of word embeddings (GPT-2, GloVe, and even arbitrary embeddings), data sets (single-subject as well as multi-subject), MEG spaces (source as well as sensor data) and type of linguistic representation (neural, artificial, or acoustic). By contrast, the second hallmark—i.e., the modulation of pre-onset encoding by next-word predictability—could only be replicated reliably in MEG in the few-subject dataset (*Figures 1C and 3C*, *Figure 1—figure supplement 1A and B*, *Figure 4—figure supplement 2A-C*), which is the same dataset analysed by *Azizpour et al., 2024*. We see two potential explanations for this discrepancy.

First, this may simply be a result of differences in the amount of data, given that the single-subject dataset comprised more than ten times the amount of data per subject than was available in the multi-subject dataset. Alternatively, and more convincingly, this discrepancy could result from a difference in the experimental designs: the multi-subject dataset did not use natural speech but carefully manipulated computer-generated speech to minimise acoustic confounds, such as co-articulation (see Methods for more details). Indeed, the second hallmark was not only absent in the neural data, but critically, also in the acoustics of the multi-subject dataset (see *Figure 3—figure supplement 1A*, *Figure 4—figure supplement 1A*). Hence, in both datasets, neural encoding closely mirrored acoustic encoding results, suggesting that ostensible hallmarks of prediction observed in the neural data reflect the correlation structure of the stimulus material rather than neural pre-activation in the brain. This apparent discrepancy, therefore, supports our proposition that stimulus dependencies constitute the driving factor behind brain encoding results.

Another important finding of the present paper is the difficulty of removing or correcting for such correlations in the stimulus material. Natural language is rife with temporal structure that is useful for predicting neighbouring words—whether such structure may be semantic, syntactic, acoustic, or due to n-gram statistics. While residualisation successfully removes dependencies within a single representational space (*Figure 4B*), brain encoding involves a mapping *between* spaces: from word embeddings to neural responses. Removing temporal correlations from but one of these two spaces still allows for the regression model to capitalise on the regularities and correlations in the second representational space. This is exemplified by our acoustic encoding analysis: even residualised word embeddings can predict the acoustics of preceding words, since the dependencies exist in language itself, not just in any particular representation of it.

Critically, we do not want to suggest that our results question the role of prediction during language processing itself. Indeed, there is a large body of work suggesting that human language processing is inherently predictive. For instance, readers and listeners are highly sensitive to even subtle fluctuations in linguistic predictability (*Smith and Levy, 2013*; *Frank et al., 2015*; *Willems et al., 2016*; *Shain et al., 2024*; *Heilbron et al., 2023*; *Brodbeck et al., 2022*; *Heilbron et al., 2022*; *Szewczyk and Federmeier, 2022*). However, such surprisal-based predictability effects on brain responses to language are usually post-stimulus (and hence indirect), while pre-stimulus evidence has, for some researchers, been considered the 'gold standard' in evidence for linguistic prediction (*Kuperberg and Jaeger, 2016*; *Pickering and Gambi, 2018*; *Nieuwland, 2019*). On first consideration, encoding modelling provides a new and more direct line of evidence that can assess pre-stimulus prediction (*Schrimpf et al., 2021*; *Goldstein et al., 2022b*; *Caucheteux et al., 2023*; *Azizpour et al., 2024*). However, due to the opacity of the word embeddings and regression models involved, interpreting these results as evidence for prediction in the brain is challenging (*Antonello and Huth, 2024*; *Azizpour et al., 2024*), rendering the evidence less direct than it may initially appear, even when it concerns evidence of pre-stimulus brain activity.

While these results, taken together, pose challenges to using pre-stimulus brain encoding to test for neural pre-activation, we yet see two possible future applications to use this analytical framework for this purpose. First, the predictability of the stimulus could be used as a threshold or benchmark: If brain encoding prior to word onset is quantitatively higher than the predictability of the stimulus, this might indicate that a predictive representation adds to the encoding performance stemming from temporal correlations alone. Although this is not the case in this current study, MEG is limited in terms of signal-to-noise ratio, and less noisy data, such as ECoG, might be able to fulfill this criterion. Second, stimulus predictability could be used as a tool in order to pre-select trials in which stimulus correlations do not allow for pre-onset encoding or favour a different trend (as in the case of our multi-subject dataset).

While our analyses focused on linear regression-based *encoding* models, the same logic applies to decoding approaches. *Goldstein et al., 2022b* also presented decoding evidence, using convolutional neural networks to decode word identity from pre-onset neural activity. However, the same fundamental ambiguity remains: If temporal dependencies enable a regression model to predict neural activity from future word embeddings, they equally enable a decoder to predict future word identity from current neural activity—as both exploit the same underlying correlations in the stimulus material. Indeed, more powerful non-linear decoders may only amplify the problem as they learn to exploit subtler dependencies that linear models might miss. The control system logic applies here as

well: If a decoder can predict upcoming words from pre-onset acoustics, this would demonstrate that decoding performance need not reflect neural pre-activation. In other words, if the brain generates predictive pre-activations, this should increase mutual information between pre-onset neural activity and the upcoming word beyond what is present in the stimulus itself.

Embedding-based encoding and decoding analyses represent an enticing new approach to studying language processing. However, their reliance on statistical prediction of brain activity, paradoxically, renders these methods difficult to use in order to test for predictions in brain activity. Since the same regularities that the brain may use to predict natural language can also be exploited by the regression model, it is ultimately difficult to know which system is performing the prediction—the brain or the regression model used by the researchers.

## Methods

### Data

To test whether previously observed evidence for encoding a word's pre-activation in neural data can be replicated in non-invasive MEG data, we used a publicly available, high-quality MEG data set (*Armeni et al., 2022*) in which three subjects listened to the audiobook version of the entire Sherlock Holmes corpus in 10 1-hr-long recording sessions. To minimise noise, head motion was restricted using individual 3D-printed head casts (for details on the data set and stimuli see *Armeni et al., 2022*). Brain responses were source localised and minimally filtered between 0.1–40 Hz. For the analysis, the MEG data was time-locked to word onset from −2000 ms to +2000 ms without applying any baseline correction. Subsequently—akin to the analysis performed by Goldstein et al.—neural activity was averaged over a sliding window of 100 ms which moved with a step size of 25 ms. This resulted in 85.719 trials of 157 time points within the window of 4s.

To examine whether our findings replicate in a more conventional multi-subject data set, we repeated our analyses in another publicly available MEG data set with 27 participants (see *Gwilliams et al., 2022* for more details). Crucially, the data set differed in several important aspects from the main data set used in this analysis. First, participants listened to four different stories within their 1 hr-long recording session, resulting in a total of 7745 trials per participant. Second, listening was interrupted by a word list or question every 3 min. Third, the speech rate varied every 5–20 sentences between 145 and 205 words per minute, and silences between sentences varied from 0–1000 ms. And lastly, MEG data was not source-localised, but all analyses were performed on channel data.

### MEG encoding modelling

The neural response to any given word was predicted based on word embeddings by means of a ridge regression—separately for each source and time point—within a 10-fold cross-validation. Within each fold, the features of the models were standard scaled before running the regression. For each time point, the predicted response was then correlated with the actual response (see *Figure 1A* for a visualisation). As features in the regression model, embeddings were obtained for each word.

For the non-contextualised analysis, we used 300-dimensional GloVe vectors (*Pennington et al., 2014*) obtained from the *spacy* package, version 3.4.3 (*Honnibal et al., 2020*). For the contextualised analysis, 768-dimensional word embeddings were extracted from the eighth layer of Hugging Face's (version 4.23.1) pre-trained GPT-2-S (*Radford et al., 2019*), as middle layers have been shown to result in the best brain encoding performance (*Goldstein et al., 2022a*; *Caucheteux and King, 2020*; *Schrimpf et al., 2020*). For words which consisted of multiple byte-pair encoded tokens (BPEs), such as 'Sherlock' which is broken down into 'S-her-lock,' the embedding of the last BPE was used. For computational reasons, the contextual window for each word ranged from 512 to 1024. For the arbitrary model, 300-dimensional word-specific vectors were drawn from a Gaussian distribution ($M = 0.1$, $SD = 1.1$). Hence, arbitrary vectors contained no systematic information other than word identity.

In order to test the second hallmark, i.e., the sensitivity of the encoding performance to the predictability of a word, we split the data into easily predictable and less predictable words, ran a separate encoding model for each split and compared their encoding performance. Since GPT-2's internal word representations are a combination of previous word representations, encoding results are difficult to interpret temporally. Consequently, this analysis was performed for non-contextualised GloVe embeddings only. In order to ascertain whether the encoding performance for predictable

words was significantly larger than for unpredictable words, we performed cluster-based permutation testing using threshold-free cluster enhancement (TFCE) with 10,000 permutations as implemented in mne stats' permutation_cluster_1samp_test function. Differences were considered significant if the computed t-value exceeded the 95th percentile under the permutation distribution.

Words were defined as easily predictable as opposed to less predictable words based on GPT-2-XL's top-one prediction. This resulted in 30,598 predicted and 55,129 unpredicted words for the few-subject dataset and in 2124 predicted and 5624 unpredicted words for the multi-subject dataset. Given that GPT-2-XL's top-one prediction might constitute a conservative estimate of whether or not a word was predictable in context, we repeated this analysis but defined words as easily predictable if they were among GPT-2-XL's top-five predicted words. This resulted in 50,666 predicted and 35,053 unpredicted words for the few-subject dataset and in 3703 predicted and 4042 unpredicted words for the multi-subject dataset. Results for these different splits are shown in the supplement.

## Source selection

Given that the purpose of this study was to encode neural responses related to linguistic predictions, we aimed to restrain our model to sources related to language processing. Hence, sources were selected for each subject individually according to a two-step procedure. First, prior to the encoding modelling, we pre-selected sources based on whether they were located in the bilateral language network (see *Heilbron et al., 2022*). This resulted in 100 sources, which were used for the encoding model. The data matrix, therefore, had the shape 100×85,719×157. Out of the resulting 100 encoded sources, we retained only those per subject which proved to allow for good encoding performance post word onset.

For this purpose, we determined for each subject the peak encoding performance in the post-word-onset window of 0–500 ms. We subsequently defined a cut-off threshold of what constitutes a 'good encoding performance' of at least 30% of that peak value. A source was then considered to allow for good encoding if it reached this threshold within the post-word-onset window of 0–500 ms. We chose this time window for our selection process, since encoding related to the word itself (as opposed to other spurious elements in the data) should be highest while the word is perceived and processed, i.e., during a time window when well-known components, such as the N400 or P600 are usually observed. Additionally, both hallmarks of prediction concern the encoding performance prior to word onset since they are supposed to reflect an encoding of the pre-activation of the representation of a word. Hence, selecting sources based on post-onset encoding performance avoids double-dipping.

Source selection was based on GloVe encoding results. This procedure resulted in 32 sources for subject 1 (max = 0.150, threshold = 0.045), 33 sources for subject 2 (max = 0.103, threshold = 0.031), 25 sources for subject 3 (max = 0.187, threshold = 0.056). To ensure that source selection was stable across neural network representations, we performed the same procedure based on GPT-2 encoding results. This resulted in fewer sources (26, 22, and 20 sources for subjects 1, 2, and 3, respectively), all of which were a subset of the sources obtained from the GloVe-based selection. Hence, all analyses were performed with the GloVe-based selection in order to ensure greater inclusivity.

Since analyses in the multi-subject data were performed on channel and not source-localised data, and since encoding was performed on group-level, channel selection was based on a simple common cut-off threshold. Hence, the encoding model was run on all 28 channels for each subject separately, resulting in 27 data matrices of the size of 208×7745×157. For each participant in the dataset, channels were retained for plotting if the channel reached the threshold in the post-word-onset window of 0–500 ms. This threshold was selected based on our results from the few-subject analysis (threshold = 0.0321). This resulted in the exclusion of subject 12 for whom no channel surpassed the threshold.

## Control system one: self-predictability analysis

As mentioned above, due to the inherent structure present in natural language, neighbouring words can share information, and therefore, neighbouring word representations can be correlated. For instance, nouns are frequently preceded by articles or prepositions, and neighbouring words belong to a similar semantic field ('pine tree,' 'driving a car,' etc.). Hence, a positive encoding performance prior to word onset might result from neighbouring embeddings being correlated, and each embedding encoding the neural representation of the corresponding word, not due to a pre-activation in the

neural signal. In other words, if a word's representation $x_0$ and the representation of the preceding word $x_{-1}$ are correlated ($\rho(x_{-1}, x_0) > 0$) and each representation ($x_{-1}, x_0$) successfully encodes its corresponding neural activity ($y_{-1}, y_0$), then encoding prior to word onset is possible since $x_0$ can be used to approximate $x_{-1}$, leading to a lower, but positive encoding performance. Word vector representation, therefore, constitutes a meaningful control system, since stimulus dependencies present in natural language are encoded within them, while they do not perform any active prediction themselves.

In order to investigate this possibility, we constructed an encoding model in which the dependent variable, i.e., the y-matrix for each trial, did not consist of neural data, but of the embedding vectors of the words that were presented at each time point. For instance, given the sentence 'You know my methods, Watson.' time-locked to the onset of the word 'methods,' we computed the onset and offset times of each of our 157 sliding time points, determined which word was presented at that time point and filled that data point with the vector for that word. For example, if we assume that each word above had a duration of 500 ms, the 20 time points between −1500 and −1000 ms were filled with the vector for 'You,' the next 20 time points with the vector for 'know' etcetera (see *Figure 2A*). Due to computational reasons, self-predictability was only computed for the first session in the few-subject dataset, i.e., approximately 10% of the available data (8622 words). We deemed this sufficient since the stimulus material in the few-subject dataset consisted of one text corpus, namely the full Sherlock Holmes corpus, and therefore, the correlational structure in 10% of the data might reasonably be representative for the whole text. Additionally, unlike the MEG data in our brain encoding model, for GPT-2, the dependent variable was a 768×8622×157 -dimensional matrix, and for GloVe and arbitrary vectors, it was 300×8622×157 -dimensional.

Akin to the brain encoding, modelling was performed by means of a 10-fold cross-validated ridge regression in order to predict previous embeddings from the embedding at time point zero and correlate the predicted and actual embeddings. This regression was performed for each feature and time point separately, and both the y- and the X-matrix were standard scaled within each fold. The resulting correlation will be referred to as the *self-predictability* of a model.

In order to test whether model self-predictability would also be able to account for the second proposed hallmark of prediction, namely, sensitivity to next word predictability, we repeated the same procedure as for the neural data. To compare the self-predictability of GloVe vectors of predictable as opposed to less predictable words, we split the data again based on GPT-2-XL's top-one prediction (see supplement for results from splits based on GPT-2-XL's top-five prediction). This split resulted in 3075 correctly and 5547 incorrectly predicted trials (5054 and 3568 for correct and incorrect predictions in the top-five split), thereby closely mirroring the percentages from the whole data set. Since our multi-subject dataset consisted of merely 7445 trials, the entirety of the data was used for the self-predictability analysis. Hence, the splits resulted in 2124 predicted and 5621 unpredicted words (for the top-one split) and in 3703 predicted and 4042 unpredicted words (for the top-five split).

In order to ensure that the results from our control analysis are not specific to the stimulus material of our two MEG datasets, but generalise to the original study that first proposed the two hallmarks of prediction tested here, we applied our control analyses also to the stimulus material used by *Goldstein et al., 2022b*. In their study, the authors presented their participants with the first 30 min of the episode *Monkey in the Middle* of the *This American Life* podcast, resulting in a total of 5136 words. Since the authors used the last layer, i.e., layer 47, of GPT-2-XL in their original study, we also used their published GPT embeddings for the analysis instead of using layer 8 of GPT-2-S. Hence, the dependent variable, the y-matrix, was a 1600×5136×157 -dimensional matrix, instead of being 768-dimensional as in the case of our few- and multi-subject datasets. The predictability split resulted in 1485 predicted and 3654 unpredicted words (for the top-one split) and in 2605 predicted and 2532 unpredicted words (for the top-five split).

## Control system two: acoustic encoding analysis

Our first control system used word embeddings (GPT, GloVe, and arbitrary representations) in order to predict word embeddings that had occurred up to 2 s prior to that word. Hence, the mapping performed by the regression model occurred within one representational system. Brain encoding designs, however, involve a mapping from one feature space, e.g., word embeddings, to an entirely different representational system, i.e., MEG or ECoG responses. In order to ascertain to

what extent temporal dependencies in the stimulus material might still be able to explain ostensible hallmarks of prediction, even when mapping between representational spaces, we analysed the predictability of the acoustics of each word. We deem word acoustics to constitute a meaningful, passive control system, as they must be represented in the brain of our listeners, and yet they in themselves quite obviously do not perform any active prediction. We obtained an acoustic embedding of each word by computing the average 8-Mel spectrogram and envelope of that word. Like for the self-predictability analysis, we constructed acoustic epochs by computing the onset and offset times of each of our 157 time points within the 4$s$ interval, determined which word was presented at that time point, and filled that data point with the acoustic representation of that word. This resulted in a y-matrix of 85,719×9 for the few-subject dataset, 7745×9 for the multi-subject dataset, and 5136×9 (*Goldstein et al., 2022b*)'s podcast dataset. We then tested the predictability of this representation of our stimulus material, and its modulation by next-word predictability using the same encoding model as used for the brain encoding and self-predictability analysis (see *Figure 3A*).

Note that for the multi-subject dataset, no information about the offset of a word was available; hence, we used the onset of the next word as a proxy for a word's offset. This, however, meant that in our multi-subject dataset, the acoustic representation for each word contained silences occurring between words and sentences. This—with no doubt—resulted in poorer acoustic embeddings and might have driven the differences in time course and the lack of a predictability effect observed in the acoustic encoding of this dataset.

Crucially, in the podcast data, the words preceding predictable or unpredictable words, i.e. the words that had to be predicted by the regression model, differed substantially in their part-of-speech (PoS) tags, with more content words for predictable trials (52% vs 45%) and more function words for unpredictable trials (55% vs 48%). Crucially, this was not the case for our few subject dataset in which content and non-content words were almost perfectly balanced between splits, at 46% and 54%, respectively. Hence, since content and non-content words differ in length and acoustic variability, and this acoustic variability of highly frequent function words has been found to depend on their predictability in context (*Bell et al., 2003*), we expected these differences to be a potential confounding factor in the encoding performance. Furthermore, we expected differences in the amount of trials between the splits to affect encoding performance, especially in low data regimes. To account for these potential confounds in the acoustics of *Goldstein et al., 2022b*'s podcast data, we randomly sub-sampled the predictable split to resemble the PoS distribution of the overall text without eliminating bigrams. We then sub-sampled the unpredictable split to resemble the resulting PoS distribution of the predictable split, while keeping the number of trials in each split of similar magnitude. This resulted in 857 predictable and 792 unpredictable trials for the top-one split, and 1554 predictable and 1480 unpredictable trials for the top-five split. We repeated this procedure for 100 random seeds to avoid artifacts caused by a specific random draw.

## Accounting for stimulus dependencies

In order to assess whether methods that have been proposed to correct for stimulus dependencies truly eliminate their influence on the hallmarks of prediction put forth by *Goldstein et al., 2022b*, we applied two possible approaches—removing reoccurring bigrams and vector residualisation—to both our control systems.

### Removal of reoccurring bigrams

The first approach for controlling for stimulus dependencies we tested was the removal of all re-occurrences of bigrams from our data. Hence, for any bigram of words occurring in the stimulus material, we only retained its first occurrence. This resulted in a reduction of data to 43,670 (from 85,719) for the few-subject dataset, to 6566 (from 7445) for the multi-subject dataset, and to 4110 (from 5136) for the podcast dataset. We re-tested for positive pre-onset encoding performance—especially focusing on our arbitrary model—within each of our two control systems. If removing all reoccurring bigrams can account for temporal dependencies in the stimulus set, using arbitrary vectors in which nothing but word identity is encoded should not result in any positive pre-onset encoding in our passive control systems.

## Residualisation of word embeddings

As mentioned above, correlations between neighbouring, structured word embeddings (such as GloVe and GPT-2 representations) might explain positive pre-onset encoding results. Since neighbouring word embeddings are positively correlated, the word embedding of the word in question ($w_0$) can be used to predict the preceding word ($w_{-1}$), and can, therefore, be used in order to predict brain activity prior to word onset without any predictive pre-activation in the neural signal. In this second approach, we, therefore, aimed to determine whether correcting for this correlation within the word embeddings is sufficient to account for the temporal dependencies in the stimulus material. For this purpose, word embeddings were residualised. For each word in the text, we regressed out all the information of the embedding at time point zero ($x_0$) that could be linearly predicted from the preceding embedding ($x_{-1}$). Hence, given the sentence 'You know my methods,' 'You' is regressed out of 'know,' 'know' is regressed out of 'my,' and 'my' out of 'methods.' This analysis can be seen as a generalised version of the control analysis performed by *Goldstein et al., 2022b*, based on directly projecting out neighbouring word embeddings. We then re-tested for positive pre-onset encoding and its modulation by predictability using residualised word embeddings in both our control systems as well as the two MEG datasets.

## Additional information

### Funding

| Funder | Grant reference number | Author |
|---|---|---|
| Nederlandse Organisatie voor Wetenschappelijk Onderzoek | VI.C.231.043 | Floris P de Lange |
| European Research Council | No. 101000942 SURPRISE | Floris P de Lange Micha Heilbron |
| European Research Council | Skłodowska-Curie grant agreement No. 945339 | Jakub Szewczyk |
| POLONEZ BIS | project No. 2022/47/P/HS6/02294 | Jakub Szewczyk |

The funders had no role in study design, data collection and interpretation, or the decision to submit the work for publication.

### Author contributions

Inés Schönmann, Conceptualization, Formal analysis, Validation, Investigation, Visualization, Methodology, Writing – original draft, Writing – review and editing; Jakub Szewczyk, Formal analysis, Supervision, Writing – review and editing; Floris P de Lange, Supervision, Funding acquisition, Investigation, Project administration, Writing – review and editing; Micha Heilbron, Conceptualization, Data curation, Formal analysis, Supervision, Investigation, Visualization, Methodology, Writing – original draft, Writing – review and editing

### Author ORCIDs

Inés Schönmann https://orcid.org/0009-0005-1915-4231
Jakub Szewczyk https://orcid.org/0000-0003-4464-082X
Floris P de Lange https://orcid.org/0000-0002-6730-1452
Micha Heilbron https://orcid.org/0000-0003-3039-4007

Reviewer #1 (Public review): https://doi.org/10.7554/eLife.106543.3.sa1
Reviewer #2 (Public review): https://doi.org/10.7554/eLife.106543.3.sa2
Reviewer #3 (Public review): https://doi.org/10.7554/eLife.106543.3.sa3
Author response https://doi.org/10.7554/eLife.106543.3.sa4

# Additional files

## Supplementary files
MDAR checklist

## Data availability
The main dataset used here, *Armeni et al., 2019*'s few-subject MEG dataset, was made available with the original publication at https://doi.org/10.1038/s41597-022-01382-7. The additional multi-subject dataset by *Gwilliams et al., 2022* is available at https://doi.org/10.17605/OSF.IO/AG3KJ. The stimuli and model features used in *Goldstein et al., 2022b* are available at https://openneuro. org/datasets/ds005574/versions/1.0.2 and the audio is available at https://www.thisamericanlife. org/631/so-a-monkey-and-a-horse-walk-into-a-bar/act-one-0. The code used for modelling analyses and plotting is available at https://github.com/InesSchoenmann/Lingpred (copy archived at *Schoenmann, 2026*).

The following previously published datasets were used:

| Author(s) | Year | Dataset title | Dataset URL | Database and Identifier |
|---|---|---|---|---|
| Armeni K, Güçlü U, van Gerven M, Schoffelen J-M | 2022 | A 10-hour within-participant magnetoencephalography narrative dataset to test models of naturalistic language comprehension | https://doi.org/10.34973/5rpw-rn92 | Donders Data Repository, 10.34973/5rpw-rn92 |
| Zada Z, Nastase SA, Aubrey B, Jalon I, Goldstein A, Michelmann S, Wang H, Hasenfratz L, Doyle W, Friedman D, Dugan P, Melloni L, Devore S, Devinsky O, Flinker A, Hasson U | 2025 | The "Podcast" ECoG dataset | https://doi.org/10.18112/openneuro.ds005574.v1.0.2 | OpenNeuro, 10.18112/openneuro.ds005574.v1.0.2 |
| Gwilliams L, Flick G, Marantz A, Pylkkänen L, Poeppel D, King JR | 2022 | MASC-MEG | https://doi.org/10.17605/OSF.IO/AG3KJ | Open Science Framework, 10.17605/OSF.IO/AG3KJ |

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
