## [Editor Report · eLife Assessment]

This **fundamental** study investigates whether neural prediction of words can be measured through pre-activation of neural network word representations in the brain; **convincing** evidence is provided that neural network representations of neighboring words are correlated in natural language. This study urges future studies to carefully differentiate between neural activity that predicts the upcoming word and neural activity that encodes the current words, which contain information that can be used to predict the upcoming word. The study is of potential interest to researchers investigating language encoding in the brain or in large language models.

---

## [Referee Report · Reviewer #1 (Public review)]

Summary:

This paper tackles an important question: What drives the predictability of pre-stimulus brain activity? The authors challenge the claim that "pre-onset" encoding effects in naturalistic language data have to reflect the brain predicting the upcoming word. They lay out an alternative explanation: because language has statistical structure and dependencies, the "pre-onset" effect might arise from these dependencies, instead of active prediction. The authors analyze two MEG datasets with naturalistic data.

Strengths:

The paper proposes a very interesting alternative hypothesis for claims in prior work (e.g., Goldstein et al., 2022). In contrast to claims in prior work, the current paper convincingly demonstrates that prior results can be explained by inherent stimulus dependencies in natural language, as opposed to the brain actively predicting future linguistic content.

Two independent datasets are analyzed. The analyses with the most and least predictive words is clever, and is nicely complementing the more naturalistic analyses. The work emphasizes how claims about linguistic prediction cannot be trivially drawn using encoding models in naturalistic designs.

---

## [Referee Report · Reviewer #2 (Public review)]

Summary:

At a high-level, the reviewers demonstrate that there is a explanation for pre-word-onset predictivity in neural responses that does not invoke a theory of predictive coding or processing. The paper does this by demonstrating that this predictivity can be explained solely as a property of the local mutual information statistics of natural language. That is, the reason that pre-word onset predictivity exist could simply boil down to the common prevalence of redundant bigram or skip-gram information in natural language.

Strengths:

The paper addresses a problem of significance and uses methods from modern NeuroAI encoding model literature to do so. The arguments, both around stimulus dependencies and the problems of residualization, are compellingly motivated and point out major holes in the reasoning behind several influential papers in the field, most notably Goldstein et al. This result, together with other papers that have pointed out other serious problems in this body of work, should provoke a reconsideration of papers from encoding model literature that have promoted predictive coding. The paper also brings to the forefront issues in extremely common methods like residualization that are good to raise for those who might be tempted to use or interpret these methods incorrectly.

Weaknesses:

After author revision, I see no major weaknesses in the underlying arguments or data processing steps.

---

## [Referee Report · Reviewer #3 (Public review)]

Summary:

The study by Schönmann et al. presents compelling analyses based on two MEG datasets, offering strong evidence that the pre-onset response observed in a highly influential study (Goldstein et al., 2022) can be attributed to stimulus dependencies-specifically, the auto-correlation in the stimuli-rather than to predictive processing in the brain. Given that both the pre-onset response and the encoding model are central to the landmark study, and that similar approaches have been adopted in several influential works, this manuscript is likely to be of high interest to the field. Overall, this study encourages more cautious interpretation of pre-onset responses in neural data, and the paper is well written and clearly structured.

Strengths:

• The authors provide clear and convincing evidence that inherent dependencies in word embeddings can lead to pre-activation of upcoming words, previously interpreted as neural predictive processing in many influential studies.

• They demonstrate that dependencies across representational domains (word embeddings and acoustic features) can explain the pre-onset response, and that these effects are not eliminated by regressing out neighboring word embeddings-an approach used in prior work.

• The study is based on two large MEG datasets and one ECoG dataset, showing that results previously observed in ECoG data can be replicated in MEG. Moreover, the stimulus dependencies appear to be consistent across the three datasets.

Weaknesses:

• While this study shows that stimulus dependency can account for pre-onset responses, it remains unclear whether this fully explains them, or whether predictive processing still plays a role. The more important question is whether pre-activation remains after accounting for these confounds.

Comments on revisions:

I appreciate the added analyses. This study raises an important methodological concern regarding an influential paper and will certainly have a high impact on our field.

---

## [Author Response]

The following is the authors’ response to the original reviews

We thank the reviewers for their constructive feedback, which has helped preparing a substantially improved manuscript. In response to concerns about the conceptual distinction between prediction and stimulus dependency, we have fundamentally restructured the paper around the notion of passive control systems. This involved rewriting the Abstract, Introduction, and large portions of the Results (~60% of text revised).

Key changes:

- New analyses on Goldstein et al. (2022) data. We demonstrate that our findings—including the insufficiency of proposed corrections—generalise to the original dataset (Figures S2B, S3B, S5C, S6B).

- Clarified novel contribution. We now make explicit that prior control analyses (residualisation, bigram removal) do not address the concern, because hallmarks persist in passive systems that cannot predict.

- Proposed criterion for future work. Pre-onset neural encoding can only count as evidence for prediction if it exceeds a passive baseline (e.g., acoustics).

We believe the revision offers a clearer, more rigorous contribution and provides a constructive framework for evaluating claims of neural prediction.

**Public Reviews:**

**Reviewer #1 (Public Review):**
Summary:This paper tackles an important question: What drives the predictability of pre-stimulus brain activity? The authors challenge the claim that "pre-onset" encoding effects in naturalistic language data have to reflect the brain predicting the upcoming word. They lay out an alternative explanation: because language has statistical structure and dependencies, the "pre-onset" effect might arise from these dependencies, instead of active prediction. The authors analyze two MEG datasets with naturalistic data.Strengths:The paper proposes a very reasonable alternative hypothesis for claims in prior work. Two independent datasets are analyzed. The analyses with the most and least predictive words are clever, and nicely complement the more naturalistic analyses.Weaknesses:I have to admit that I have a hard time understanding one conceptual aspect of the work, and a few technical aspects of the analyses are unclear to me. Conceptually, I am not clear on why stimulus dependencies need to be different from those of prediction. Yes, it is true that actively predicting an upcoming word is different from just letting the regression model pick up on stimulus dependencies, but given that humans are statistical learners, we also just pick up on stimulus dependencies, and is that different from prediction? Isn't that in some way, the definition of prediction (sensitivity to stimulus dependencies, and anticipating the most likely upcoming input(s))?

We thank the reviewer for this comment, which highlights that the previous version wasn’t sufficiently clear. Conceptually, the difference is critical: it is the difference between passively encoding or representing the stimulus (like e.g., a spectrogram of the stimulus would), and actively generating predictions.

We have substantially changed the framing of the paper to put the notion of control systems centre-stage. One such control system is the speech acoustics: they encode the stimulus (and thus its dependencies) but cannot predict. When we observe the "hallmarks of prediction" in acoustics, this demonstrates the hallmarks can arise without any prediction.

This brings me to some of the technical points: If the encoding regression model is learning one set of regression weights, how can those reflect stimulus dependencies (or am I misunderstanding which weights are learned)? Would it help to fit regression models on for instance, every second word or something (that should get rid of stimulus dependencies, but still allow to test whether the model predicts brain activity associated with words)? Or does that miss the point? I am a bit unclear as to what the actual "problem" with the encoding model analyses is, and how the stimulus dependency bias would be evident. It would be very helpful if the authors could spell out, more explicitly, the precise predictions of how the bias would be present in the encoding model.

Different weights are estimated per time point in the time-resolved regression. This allows the model to learn how the response to words unfolds, but also to learn different stimulus dependencies at each timepoint. Fitting on every second word would reduce but not eliminate the problem. Our control system approach provides a more principled test. We have clarified the mechanism in the Introduction (lines 82-90), explaining how correlations between neighbouring words allow the regression model to predict prior neural activity without assuming pre-activation.

**Reviewer #2 (Public Review):**
Summary:At a high level, the reviewers demonstrate that there is an explanation for pre-word-onset predictivity in neural responses that does not invoke a theory of predictive coding or processing. The paper does this by demonstrating that this predictivity can be explained solely as a property of the local mutual information statistics of natural language. That is, the reason that pre-word onset predictivity exists could simply boil down to the common prevalence of redundant bigram or skip-gram information in natural language.Strengths:The paper addresses a problem of significance and uses methods from modern NeuroAI encoding model literature to do so. The arguments, both around stimulus dependencies and the problems of residualization, are compellingly motivated and point out major holes in the reasoning behind several influential papers in the field, most notably Goldstein et al. This result, together with other papers that have pointed out other serious problems in this body of work, should provoke a reconsideration of papers from encoding model literature that have promoted predictive coding. The paper also brings to the forefront issues in extremely common methods like residualization that are good to raise for those who might be tempted to use or interpret these methods incorrectly.Weaknesses:The authors don't completely settle the problem of whether pre-word onset predictivity is entirely explainable by stimulus dependencies, instead opting to show why naive attempts at resolving this problem (like residualization) don't work. The paper could certainly be better if the authors had managed to fully punch a hole in this.

We thank the reviewer for their assessment.

We believe our paper does punch the hole that can be punched, which is a hole in the method. Our control demonstrates that adjusting the features (X matrix) cannot address dependencies that persist in the signal itself (Y matrix). Because the hallmarks emerge in a system that cannot predict (even after linearly removing the previous stimulus) attributing pre-onset encoding performance to neural prediction (rather than stimulus structure) is fundamentally ambiguous, and different (e.g. variance partitioning) approaches would suffer from the same ambiguity. We have reframed the manuscript to make this argument more clearly.

**Reviewer #3 (Public Review):**
Summary:The study by Schönmann et al. presents compelling analyses based on two MEG datasets, offering strong evidence that the pre-onset response observed in a highly influential study (Goldstein et al., 2022) can be attributed to stimulus dependencies, specifically, the auto-correlation in the stimuli—rather than to predictive processing in the brain. Given that both the pre-onset response and the encoding model are central to the landmark study, and that similar approaches have been adopted in several influential works, this manuscript is likely to be of high interest to the field. Overall, this study encourages more cautious interpretation of pre-onset responses in neural data, and the paper is well written and clearly structured.Strengths:(1) The authors provide clear and convincing evidence that inherent dependencies in word embeddings can lead to pre-activation of upcoming words, previously interpreted as neural predictive processing in many influential studies.(2) They demonstrate that dependencies across representational domains (word embeddings and acoustic features) can explain the pre-onset response, and that these effects are not eliminated by regressing out neighboring word embeddings - an approach used in prior work.(3) The study is based on two large MEG datasets, showing that results previously observed in ECoG data can be replicated in MEG. Moreover, the stimulus dependencies appear to be consistent across the two datasets.

We’d like to thank the reviewer for their comments on our preprint.

Weaknesses:(1) To allow a more direct comparison with Goldstein et al., the authors could consider using their publicly available dataset.

We thank the reviewer for this suggestion. The Goldstein dataset was not publicly available when we conducted this research. However, we have now applied our control analyses to their stimulus material, and found that the exact same problem applies to their dataset, too.

We have added analyses of the Goldstein et al. (2022) podcast stimulus throughout the paper. Results are shown in Figures S2B, S3B, S5C, and S6B. Critically, we observe the same pattern: both hallmarks emerge in the acoustic control system, and residualisation fails to eliminate them. This demonstrates that our findings generalise to the very dataset used to establish pre-onset encoding as evidence for neural prediction.

(2) Goldstein et al. already addressed embedding dependencies and showed that their main results hold after regressing out the embedding dependencies. This may lessen the impact of the concerns about self-dependency raised here.

We thank the reviewer for raising this point, as it reveals we failed to convey a central argument in the previous version. Goldstein et al.'s control analysis did not address the concern. We show that even after the control analyses that Goldstein et al. perform (removing bigrams, regressing out embedding dependencies) the "hallmarks of prediction" still emerge when applying the analysis to a passive control system that by definition does not predict: the speech acoustics. We now also show this in their data.

To better convey this critical point, around the concept of "passive control systems". We now first establish that the hallmarks appear in acoustics (Figure 3), then show that residualisation fails to remove them (Figure 4). This makes explicit that any claim about "controlling for dependencies" must be validated against a system that cannot predict.

(3) While this study shows that stimulus dependency can account for pre-onset responses, it remains unclear whether this fully explains them, or whether predictive processing still plays a role. The more important question is whether pre-activation remains after accounting for these confounds.

We thank the reviewer for this question, and we agree that the question whether pre-activation occurs is an important and interesting one. However, we ask a different question in our study: Our goal is not to definitively establish whether the brain predicts during language processing; it is to scrutinise what counts as evidence for prediction, and to correct for some highly influential claims made in the literature. The reviewer asks whether pre-activation remains "after accounting for these confounds." But the point we are trying to make is that in this analytical framework, one cannot analytically account for these confounds: corrections to the X matrix leave dependencies in the data itself intact, as the acoustic control demonstrates.

We do offer recommendations for future work. The passive control systems approach can serve as a benchmark: pre-onset neural encoding (or decoding) can only count as evidence for prediction if it exceeds what is observed in a passive control system like acoustics (which is not what we observe). Additionally, the field could move toward less naturalistic stimuli with tighter experimental controls, reducing the correlations that make this attribution so difficult. Developing a new definitive test is beyond the scope of our paper, but we believe applying this benchmark is a necessary first step.

To make this clearer, we have rewritten the Discussion to explicitly state this criterion (lines 331-340) and to outline these recommendations for future work (lines 337-340). We have also added a paragraph extending our argument to decoding approaches (lines 343-354), noting that the same ambiguity applies regardless of analytical direction.

**Recommendations for Authors:**

**Reviewer #1 (Recommendations for Authors):**
As per my "Weakness" point, I would appreciate engagement with the conceptual point related to the difference between prediction and stimulus correlations. Most importantly, I hope the authors will spell out more explicitly which predictions their proposal makes, and how exactly those would be present in an encoding model.

Our proposal makes a clear prediction: if pre-onset encoding can be explained by stimulus dependencies (essentially a confound in the analysis) the same hallmarks should emerge in passive control systems that encode the stimulus but do not predict. We test this with word embeddings and speech acoustics, and both show hallmarks despite not doing any prediction.

**Reviewer #2 (Recommendations for Authors):**
I greatly enjoyed reading the paper and only have minor quibbles. The work is overdue and will no doubt be a valuable addition to the literature to push back on over-hyped claims about the implications of pre-word predictivity in neural response. I have few issues with the methods that the paper uses, they seem sensible and in line with previous work that has investigated these questions, and I did not find typos.One point I would like to raise is whether or not there is a more effective solution to resolving the issues behind residualization that the paper demonstrates. The authors show that removing next-word information does not effectively resolve the problem that local relationships in the stimulus dataset pose. The challenge to me here seems to be that it is difficult to get a model to "not learn" a relationship that is learnable. I wonder if a better solution to this is to not try to get a model to exclude a set of information but instead to do some sort of variance partitioning where you train a model to predict the next-word representation from the current-word representation (as in the self-predictivity analysis) and then build an encoding model out of the predicted representation. Then, compare the pre-word-onset encoding performance of the prediction with the pre-word-onset encoding performance of the original representation. If the performance of the two models roughly matches, that would be strong evidence that most of what these models are capturing before word onset is just explainable by the stimulus dependencies, no?

We would like to thank the reviewer for their kind words and positive appraisal!

The proposed analysis is that if a linear proxy representation, w_hat_t – predicted linearly from w_{t-1} – yields pre-onset predictivity comparable to the actual w_t vector, this would support that the effect can be explained by stimulus dependencies. While this is an interesting alternative analysis, we would be cautious about the inverse conclusion: that if w_t outperforms the linear proxy w_hat_t, the residual variance must reflect true neural prediction.

This is because of our control system results. We show that even when we remove the "predictable" shared variance – which is similar to computing the difference between w_t and w_hat_t – the unique information still yields pre-onset predictivity, albeit reduced, in the passive acoustics that by definition cannot predict. Therefore, instead of developing an ever-more-clever way to "correct" for the problem by adjusting the X matrix, we focus on showing that the problem lies in the stimulus itself. For the revision, we focused on reframing the problem and hope we have punched a fuller hole in the logic by breaking down the fundamental issue more clearly and showing it applies to the stimulus material of Goldstein et al. (2022) as well.

Additionally, I would say that I was a bit confused about what was going on in the methods figures, to the point where I do not see the value in having them, but thankfully, the text was clear enough to resolve that confusion.

We are sad the methods illustration wasn’t helpful. In presentations we have found that the illustrations were generally helpful to bring the analysis across, e.g. the aspect of keeping the analysis identical but simply replacing the brain data with either word vectors (current Figure 2) and acoustics (current Figure 3). In the revision we have reorganised the schematics slightly, we introduce the acoustics as a control system earlier, to separately introduce residualisation and its insufficiency (Figure 4). We hope this helps

**Reviewer #3 (Recommendations for Authors):**
(1) My major concern is the extent to which this study offers new insights beyond what was already demonstrated in Goldstein's work. First, the embedding dependency highlighted by the authors seems somewhat expected, given how these embeddings are constructed: GloVe embeddings are based on word co-occurrence statistics, and GPT embeddings are combinations of embeddings of preceding words. More importantly, Goldstein et al. addressed this issue by regressing out neighboring word embeddings. This control was effective, as also confirmed by the current manuscript, and their main results remain. Therefore, the embedding dependency appears to have been properly accounted for in the earlier study.Building on the previous point, I appreciate the analysis of dependencies across representational domains, which I see as the main novel contribution of this manuscript. I would encourage the authors to explore this aspect more deeply. If I understand correctly, stimulus dependencies may persist even after regressing out neighboring word embeddings due to two potential factors:(a) Temporal dependencies in embeddings: since the regression of neighbor words is performed at the word level rather than over time, temporal dependency may remain.(b) Cross-feature dependencies - specifically, correlations between embeddings and acoustic features.Regarding the first factor, it is not entirely clear to me whether this is a real problem—i.e., whether word-level regression fails to remove temporal dependencies. A simulation could help clarify this and support the argument. While it's not essential, it would be valuable if the authors could propose a method to address this issue, or at least outline it as a direction for future work.For the second point, it would be helpful for the authors to explicitly explain the potential relationship between word embeddings and acoustic features. Additionally, while correlations between features are a common problem in speech research, they are typically addressed by regressing out acoustic features early in the analysis (Gwilliams et al., 2022). It would strengthen the current findings if the authors could test whether the self-predictability persists even after controlling for neighboring embeddings and acoustic features.

We appreciate the extensive and detailed engagement with our work, which has been very useful in highlighting key unclarities and gaps we had to address.

We do believe our study goes well beyond what was shown by Goldstein, by identifying a fundamental limitation in their analysis, and showing that their purported control analyses do not in fact control for the problem. We’ll address the reviewers' sub-questions in turn.

(i) Why this offers crucial insights beyond Goldstein et al.

While Goldstein et al. indeed addressed embedding dependencies via residualization (or in their case projection), their conclusion relied on the assumption that any neural encoding surviving this "fix" must reflect genuine predictive pre-activation. Our study invalidates this assumption. By applying the residualization fix, we show that the "hallmarks of prediction" persist just as robustly in a passive control system that cannot predict (the speech acoustics) as in the neural data. (We also show this for bigram removal.)

This provides a key new insight: persistent pre-onset predictivity after “correction” is not evidence that the dependency issue was solved. Instead, because the same effect persists in a system that cannot predict (acoustics), the persistence of the hallmarks cannot be attributed to prediction. It demonstrates that the standard "fix" is mathematically insufficient to remove the confound, rendering the original evidence for neural prediction fundamentally ambiguous.

(ii) Why do dependencies/hallmarks persist after residualization?

Residualization successfully removes the linear dependency between the current embedding (w_t) and the previous embedding (w_{t-1}) within the feature space. However, it does not (and cannot) remove the dependency from language itself, and therefore from the brain which (in some format) encodes the linguistic stimulus. Language is massively redundant. Knowing the current word tells you something about what came before – acoustically, syntactically, semantically. As long as the embedding identifies the word, the regression model will re-learn this relationship. For instance, in the case of acoustics, even when using the corrected embedding, the regression will re-learn that certain words (e.g., "Holmes") tend to follow certain acoustic patterns (e.g., the acoustics of "Sherlock"). “This shows that correcting the embeddings is insufficient: the dependencies exist in language itself, and the model will re-learn them from any signal that encodes that language.”

(iii) Why not regress out the acoustics?

This is also why "regressing out acoustics" (as the reviewer suggests) would miss the point. We do not claim that acoustic features leak into the neural signal or that acoustics are a specific confound to be removed. Rather, we use acoustics as a “passive baseline”: a system that encodes the stimulus but cannot predict. That the method yields "hallmarks of prediction" in this baseline demonstrates these hallmarks are not valid evidence for prediction—regardless of what additional features one regresses out. This motivates our proposed criterion: future studies seeing evidence for neural pre-activation should not rest on finding pre-onset encoding per se, since passive systems show this too. Rather, it should require demonstrating that the brain signal contains more information about the upcoming word than the passive stimulus baseline.

As these aspects are fundamental to the interpretation of our study, we have fundamentally re-organised and re-wrote large parts of the paper. We hope it is much clearer now.

(2) To better compare to Goldstein's work, the author may consider performing the same analyses using their publicly available dataset.

This is a good suggestion. When we initially conducted this research, the Goldstein dataset was not yet publicly available. It now is, and we have applied our analyses to their stimulus material. The same problem emerges: the hallmarks of prediction appear in the acoustics of their podcast stimuli. Even after applying the control analyses, pre-onset predictivity is robust in their acoustics (indeed, in correlation terms, higher than reported for the neural data, so there is not more predictivity in the brain than in the stimulus material), confirming that the issue we identify applies to the original dataset. Results are shown in Figures S2B, S3B, S5C, and S6B.

(3) It is also interesting to show the predictability effect after word onsets for self-predictability analyses, for example, in Figure 2C. The predictability effect is not only reflected in pre-onset responses but also in post-onset responses, i.e., larger responses for unpredicted words. Whether the stimulus dependency mirror this effect?

Our paper focuses specifically on temporal dependencies – the capacity of the current word to predict the previous stimulus signal (e.g., previous acoustics, previous embeddings) – and how this mimics neural pre-activation. Post-onset analyses, by contrast, concerns the mapping between the current word and its concurrent signal, which involves fundamentally different mechanisms (e.g., mapping fidelity, frequency effects, acoustic clarity, word length) and would require the consideration of covariates of the attributes of the word post-onset to meaningfully interpret. Post-onset, there can be differences between predictable and non predictable words – e.g. sometimes unpredictable words are pronounced with more emphasis – which is why surprisal studies include a large range of covariates. However, this is not about stimulus dependencies or pre-activation, so we consider it is beyond scope of our study.

(4) The authors might consider reporting the encoding performance for the residual word embeddings, similar to Figure S6B in Goldstein's paper. This would allow us to determine whether pre-activation persists in the MEG responses and compare its pattern with the predictability of pre-onset acoustics.

We do report this analysis, in the revised supplement it is shown in Figure S7. We placed it in the supplement precisely because residualized embeddings are not the "fix" they appear to be: as we show, they still yield strong pre-onset predictivity in the passive acoustic baseline (Figure 4, S6), undermining their use as a control.

(5) The series of previous pre-activation analyses proposed fruitful findings, e.g., the difference between brain regions (Fig. S4, (Goldstein et al., 2022)) and the difference between listeners and speakers (Figure 2, (Zada et al., 2024)). Whether these observed differences can be explained by the stimulus dependency?

We appreciate this question. Our goal is to address the general logic of using pre-onset encoding as evidence for prediction, rather than to critique every finding in specific papers, especially as it pertains to a specific author. But briefly:

Speaker vs. Listener differences (Zada et al., 2024): Zada et al. report distinct temporal profiles: speaker encoding peaks pre-onset (planning?), whereas listener encoding peaks post-onset but shows a pre-onset "ramp." Our critique applies to interpreting this ramp as "prediction." However, this interpretation is not central to their paper, which focuses on speaker-listener coupling via shared embedding spaces. We leave the implications (which are clear enough) to the reader.

Regional differences (Goldstein et al., 2022): Encoding timecourses do vary across electrodes, as we also observe across MEG sources (and participants). But our point is logical: because pre-onset encoding does not necessarily reflect prediction, finding a channel with stronger pre-onset encoding does not mean that channel performs “more prediction”. For instance, one subject in the Armeni dataset showed higher pre-onset than post-onset encoding (and indeed activity) overall – but it would be implausible to conclude this subject "only predicts" and does not “process” or “listen”. More likely, this reflects differences in signal-to-noise, integration windows, or source contributions. The exact sources of these morphological differences are interesting but unclear, and speculating on them is beyond our scope.

(6) I appreciate that the authors have shared their code; however, some parts appear to be missing. For example, the script encoding_analysis.py only includes package-loading code.

Thank you for noticing, we have updated our code database.

(7) What do the error bars in the figures represent - for example, in Figure 1C? How many samples were included in the significance tests? The difference between the two curves appears small, yet it is reported as significant. Additionally, Figure S1 shows large differences between subjects and between the two MEG datasets. Do the authors have any explanation for these differences?

The shaded areas in our previous (Figure 1c) show 95% confidence intervals computed over the 100 MEG sources identified to be part of the bilateral language system and the 10 cross-validation splits.

We do not have an elaborate explanation for the differences in encoding performance across the three subjects in the few-subject dataset. Instead, we interpret these differences as a likely consequence of substantial inter-individual variability in evoked responses, even at the source level, arising from differences in cortical folding and the orientation of underlying current dipoles. We deem this a likely explanation since different electrodes in Goldstein’s ECoG data also showed very different encoding profiles.

With respect to the multi-subject dataset, we suspect that the large differences stem most likely from two substantial differences: First, the acoustics were purposefully manipulated by the experimenters to reduce temporal dependence. This made it harder for listeners to concentrate on the stories and thereby might have potentially led to lower quality neural data. Furthermore, it reduced one form of stimulus dependency, namely the acoustic temporal dependencies, which could be exploited by the encoding model to reach higher encoding accuracies. Secondly, MEG has a notoriously poor signal-to-noise ratio, and the amount of data per participant (7.745 words as opposed to 85.719 in the few-subject dataset) might not have been enough to produce reliably high encoding results.

Finally, the current study is clear and convincing, and my suggestions are not intended to question its novelty or robustness. Rather, I believe the authors are in a strong position to address a critical question in language processing: whether pre-activation occurs. The authors have thoughtfully considered important confounds related to pre-onset responses. Adding some approaches to regressing out these confounds could be particularly helpful for determining whether a true pre-onset response remains.

We thank the reviewer again for their constructive feedback, suggestions and questions. To clarify, however, our goal is *not* to definitively attest to whether pre-activation occurs. Our goal is simply to scrutinise a specific method to test for linguistic prediction. This method purports to be an improvement on conventional post-onset (e.g. surprisal-based) methods, as it can directly investigate effects occurring prior to word onset. We have demonstrated fundamental limitations in the underlying logic of this method. We propose passive control systems as baselines against which claims of prediction should be evaluated. Against this baseline, the current evidence does not show unequivocal support for prediction: pre-onset encoding in the brain does not exceed that in the passive control. However, we do not conclude from this that pre-activation does not exist — that would require a different study entirely. Our aim is more methodological: to establish what should count as evidence for prediction, not to settle whether prediction occurs.

We would like to thank the reviewers and editors for their thoughtful feedback, which has been tremendously helpful in improving the paper.